# LBNP: Learning features between neighboring points for point cloud classification

Lei Wang[1,2], Ming Huang[2]*, Zhenqing Yang[3], Rui Wu[1,2], Dashi Qiu[2], Xingxing Xiao[1,2], Dong Li[1,2], Cai Chen[1,2]

**1** School of Architecture and Urban Planning, Beijing University of Civil Engineering and Architecture, Beijing, China, **2** School of Geomatics and Urban Spatial Informatics, Beijing University of Civil Engineering and Architecture, Beijing, China, **3** Beijing Construction Engineering Group, Beijing, China

* huangming@bucea.edu.cn

## Abstract

Inspired by classical works, when constructing local relationships in point clouds, there is always a geometric description of the central point and its neighboring points. However, the basic geometric representation of the central point and its neighborhood is insufficient. Drawing inspiration from local binary pattern algorithms used in image processing, we propose a novel method for representing point cloud neighborhoods, which we call Point Cloud Local Auxiliary Block (PLAB). This module explores useful neighborhood features by learning the relationships between neighboring points, thereby enhancing the learning capability of the model. In addition, we propose a pure Transformer structure that takes into account both local and global features, called Dual Attention Layer (DAL), which enables the network to learn valuable global features as well as local features in the aggregated feature space. Experimental results show that our method performs well on both coarse- and fine-grained point cloud datasets. We will publish the code and all experimental training logs on GitHub.

## Introduction

Point cloud classification is one of the core tasks in the field of computer vision and plays a vital role in various fields, such as urban information modeling, 3D map creation, and automatic driving.

For irregular point cloud data, PointNet [1] pioneered point-based deep neural network technology using a point-wise Multi-Layer Perceptron(MLP) and symmetric function. Subsequently, PointNet++ [2] introduced local SetAbstraction to realize local information aggregation. The development of these techniques has inspired subsequent research, and many excellent studies have developed novel and complex local aggregation modules to enhance the local information representation ability of point clouds. However, these methods mostly establish the relationship between neighborhood points and the center point to form an implicit or explicit local geometric description, thereby enhancing the aggregation and representation capabilities of the local point cloud. We believe that strengthening the correlation between neighborhood points, rather than just the correlation between neighborhood points and the

**Data Availability Statement:** The ModelNet40 dataset are available from the Kaggle database (https://www.kaggle.com/datasets/balraj98/modelnet40-princeton-3d-object-dataset) The FG3D dataset are available from Kaggle database

([https://www.kaggle.com/datasets/yue123y/fine-grained-3d](https://www.kaggle.com/datasets/yue123y/fine-grained-3d)).

**Funding:** This study is sponsored by the BUCEA Doctor Graduate Scientific Research Ability Improvement Project(DG2024034) and National Natural Science Foundation of China (42171416). The funders had no role in study design, data collection and analysis, decision to publish, or preparation of the manuscript.

**Competing interests:** The authors have declared that no competing interests exist.

center point, will be more helpful for the local feature extraction of point clouds, which will bring beneficial effects to downstream tasks.

In this paper, we propose a plug-and-play Local Auxiliary Block called the Point Cloud Local Auxiliary Block (PLAB), which focuses on enhancing the correlation between neighboring points to improve the local feature representation capabilities of point clouds. Inspired by the extraction methods of image texture features, particularly the Local Binary Pattern (LBP) [3], this method emphasizes the subtle structural differences between different regions in an image and is mainly used to describe the surface properties of an image or image region, such as the thickness and density of the texture, which aid in image discrimination. Inspired by this, we sought to use neural networks to identify potential "texture features" in point clouds to improve the expression of local features. Mimicking LBP, we propose a local feature aggregation module, which uses relationship mapping between neighboring points relative to the central point to obtain sufficient neighborhood feature information by feature aggregation instead of using an encoding process. To improve the sensitivity of neighborhood points, we transform the Cartesian coordinate system into a spherical coordinate system as a supplementary feature, and we use the spherical coordinates to sort the adjacent points clarifying the point–pair relationships. For more consistent order independence, we employ a symmetric function to aggregate the information of neighboring points.

Furthermore, we note that the over-subdivision of the PLAB structure may introduce additional noise. To address this problem, we propose a novel local and global fusion pure Transformer Layer called the Dual-Attention Layer (DAL) as our backbone network. It is well known that both local and global features play important roles in point cloud tasks. Global features can capture the shape and structural information of the point cloud, while local features provide set and detail information. Recently, various model have emerged for point cloud classification. The Transformer has received extensive attention owing to its strong long-range dependence modeling ability, and many outstanding works have emerged from it. For example, the local attention mechanism of Point Transformer [4] and global attention mechanism of Point Cloud Transformer [5] have inspired related research. Other 3D Transformer-related variants have also been successfully explored in terms of speed, accuracy, and efficiency; however, few pure Transformer models are based on both local and global information. In our Transformer module, offset-attention is used as the global Transformer module, and the calculated global attention scores is introduced into the local attention module with high attention position indices, forming the DAL.

We propose a complementary module for local neighborhoods and a pure Transformer layer model designed for local and global modeling in point cloud classification tasks.

Our main contributions are as follows:

·A novel local aggregation module, PLAB, is proposed, which can improve all types of point cloud networks with local aggregation properties to different degrees.

·The PLAB module is plug-and-play. Without changing the original network structure, it can be inserted directly into the model to improve its effectiveness.

·A new pure Transformer structure that integrates global and local features, DAL, is characterized by low parameter count and superior performance, showing promising results on classification datasets.

## Related work

### Deep learning for point clouds

Owing to the disordered and unstructured nature of point clouds, early deep learning methods were usually based on multi-view or voxel methods. Multi-view methods [6–8] usually use

multiple viewpoints of the point cloud to capture richer information to improve task performance. However, this projection method cannot adapt to surface density changes and is affected by occlusion. Voxel methods [9–13] map 3D space to a regular voxel grid, so 3D convolution can be used naturally, but such methods are prone to losing fine-grained geometric features, and the voxelization process results in heavy memory and computational consumption. PointNet pioneered a point-based approach that uses a shared MLP to encode each individual point and global pooling to aggregate each encoded point feature. PointNet++ is an extension of PointNet, by using multi-scale local PointNets to enhance the geometric representation. Numerous other studies have also built upon this foundational works.

## Point cloud neighborhood feature aggregation

Generally, local aggregation operators can be divided into two categories: convolution- and graph-based methods. Among the convolution-based methods, PointCNN [14] uses x-transformation to learn the weights and order of neighborhood points to ensure that the point cloud can perform convolution operations. KPConv [15] associates the weight matrix with a predefined kernel point in 3D space to adapt it to the uneven distribution of local points. PAConv [16] constructs a position-adaptive convolution operator with a dynamic kernel, which combines weight matrices in a weight library, and utilizes MLPs to learn weight coefficients based on relative point positions. Among graph-based methods, DGCNN [17], as a pioneering method, aggregates neighboring points in the feature space on a dynamically updated graph at each layer. CurveNet [18] groups line segments connected into lines in a point cloud expanding the receptive field and explore useful geometric information. Most of these methods adhere to the basic paradigm of PointNet++, wherein the relationship between adjacent points and center points are constructed by absolute or relative coordinates, and then other simple or complex operations are used to obtain more explicit local geometric features.

In previous approaches, a widely used encoding approach is by constructing the coordinates of neighboring points relative to the center point, which brings benefits to the model in terms of position invariance and preservation of local structure [14–17]. However, we believe that such an encoding approach ignores the associations between neighboring points, because the local geometry exists not only in the relative relationship between neighboring points and the centroid, but as a whole in all pairs of points. In order to enhance the associations between point pairs, we attempt to enhance the representation of geometric information in the neighborhood by exploring the relationships between neighborhood points used to aggregate features, hence the proposed PLAB module.

## Point cloud transformer

Transformers have made significant progress in the field of computer vision, and a series of explorations have been conducted in the 3D point cloud field. PCT is a classical global Transformer structure that first employs neighborhood embedding to aggregate local information and then uses this information as a four-layer stacked global Transformer block. Finally, global Max pooling and average pooling are applied to obtain the global classification information. 3CROSSNet [19] utilizes cross-level, cross-scale, and cross-attention strategies to capture point cloud features. The network first obtains point cloud subset features at different resolutions through FPS and MLP and then uses the stack and fusion of Transformer blocks to obtain complex structures and details in the point cloud. Referring to the Canny edge detection operator, APES [20] explored a point cloud downsampling method combined with an attention mechanism to capture salient points in the point cloud contour. Specifically, this method calculates the correlation between points and selects the points with high correlation, namely, the

salient points on the contour for sampling. Thus, the network can learn and extract edge features from the data more effectively. The idea of the Stratified Transformer [21] is similar to that of Swin Transformer [22], which divides the point cloud into different windows to reduce the computation of the Transformer. To achieve information interaction between different windows and improve the receptive field, window displacement operation and key sampling stratification strategy are adopted.PT [4] is another classic local Transformer structure composed of five layers of continuously downsampled local Transformer blocks. For each block, it uses KNN to obtain nearby points, and for each neighborhood point set, it uses the vector attention mechanism to capture local features. PTv2 [23] also introduced group vector attention, position encoding multipliers, and pooling based on segmentation to enhance local geometric capture and ensure sensitivity to global context. Subsequent PTv3 [24] introduced serialized encoding to achieve faster and better results. PointConT [25] uses the locality of points in the feature space to cluster sampling points with similar features into the same class and calculates the self-attention for the points in each class. In the feature aggregation of PointConT, the feature vector is divided into two branches of low and high frequency through maximum and average pooling, respectively, which are used to extract local information more fully. In the above studies, the global attention mechanism fails to provide finer local geometric capture [5, 19, 20], while the local attention mechanism loses its long-range dependency [4], and the hybrid approach incorporates other complex modules or structures [21, 23–26]. However, it is easy to be overlooked that the global attention matrix is the neighborhood representation of points in the feature space, and we propose that DAL using the neighborhood features indexed by the attention graph can preserve the long-range dependence property inside the Transformer while accomplishing the local aggregation.

## Methodology

### Overview

Fig 1 shows our network architecture, in which the PLAB structure is utilized in the Embedding layer to assist local feature extraction, and the main framework is constructed by stacking the DAL, which is composed of Offset-Attention and External-Attention [27]. Specifically, in

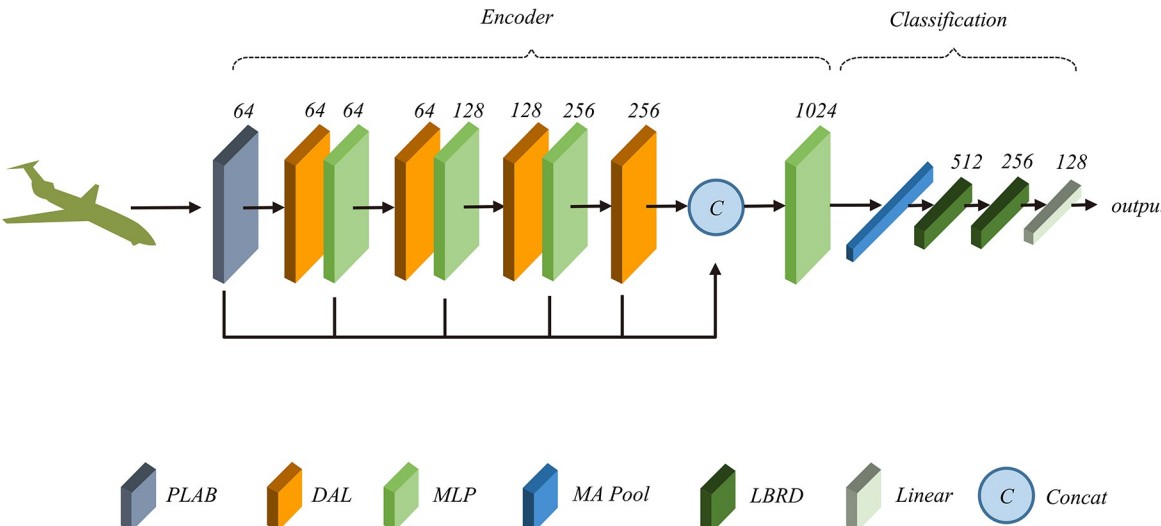

**Fig 1. Model structure.** The encoder part consists of a layer of PLAB to expand the dimension, four layers of DAL with MLP to mine features, MA Pooling to connect *Max Pool* and *Average Pool*, and LBRD consisting of *Linear*, *BatchNorm*, *ReLU*, and *Dropout*.

the PLAB layer, point cloud coordinates are used as the main features, and the initial features are projected into a higher dimension space using a local MLP and symmetric function. PLAB is used as an auxiliary feature and is extended to the same dimension as the main feature by MLP and the symmetric function and then fused with it. The k-nearest neighbor point clusters are organized and sorted according to the spherical coordinates to establish the point–pair relationship among the neighborhood points and calculate the relative coordinates. The transformed spherical coordinates are added to the neighborhood point pairs as supplementary features. We want the features of the neighborhood pairs to be optional; therefore, we employ Attentive Pooling [28] as the aggregation unit which assign different weights to the feature channels and finally aggregate them using the summation function.

The global module of DAL utilizes the dot product of the query vector, key vector, and *softmax* to compute the attention score map, which serves as a feature selector. Specifically, it has been demonstrated that most pixels are closely related to only a few other pixels [27], and this is also applicable to point clouds. Therefore, the attention score formed by the dot product can select the Top-K most relevant to the current point to fully describe the feature information of the point. The selected Top-K feature vectors are then sent to the local Transformer module for local attention calculation. However, due to the quadratic complexity of the attention calculation incurs huge computational overhead, if the local attention calculation still uses the dot product attention form will become DAL huge. Therefore, we utilize the lightweight attention mechanism External-Attention for local attention calculation. This not only has the lightweight advantage of linear complexity but also the memory unit can pay attention to the potential relationship between different local features.

## Point cloud local auxiliary block

Our PLAB module is based on the texture feature extraction algorithm, LBP, which is an operator designed the local texture features of an image and can be employed for image feature analysis. Here, we provide a briefly introduce the classical LBP operator. The classical LBP operator is defined as a 3×3 square window, where the center pixel of the window is taken as the threshold and the gray value of its eight neighboring pixels is compared with the pixel value of the center of the current window. If the pixel value of the neighborhood is less than that of the center point, the value of the pixel is set to 0; otherwise, the value of the pixel is set to 1. In this manner, the eight pixels in the neighborhood of a 3×3 window are compared with the center pixel, and an 8-bit binary number is generated, which can generate 256 LBP codes. The LBP codes calculated using this method can be used to reflect the regional texture feature information of the window.

We note that the size comparison of neighborhood points relative to the center point within the local window can be interpreted as a way to establish the relationship between the neighborhood and center point, such as their relative position in the point cloud, whereas the binary encoding process can be considered as a process to establish the relationship between neighborhood points. As show in Fig 2, we describe this process as a simple deep learning task for point clouds:

1. obtain the nearest neighbors of the point cloud;

2. sort the neighboring points to establish the relationship between the neighboring point pairs;

3. learn feature information about the neighboring points.

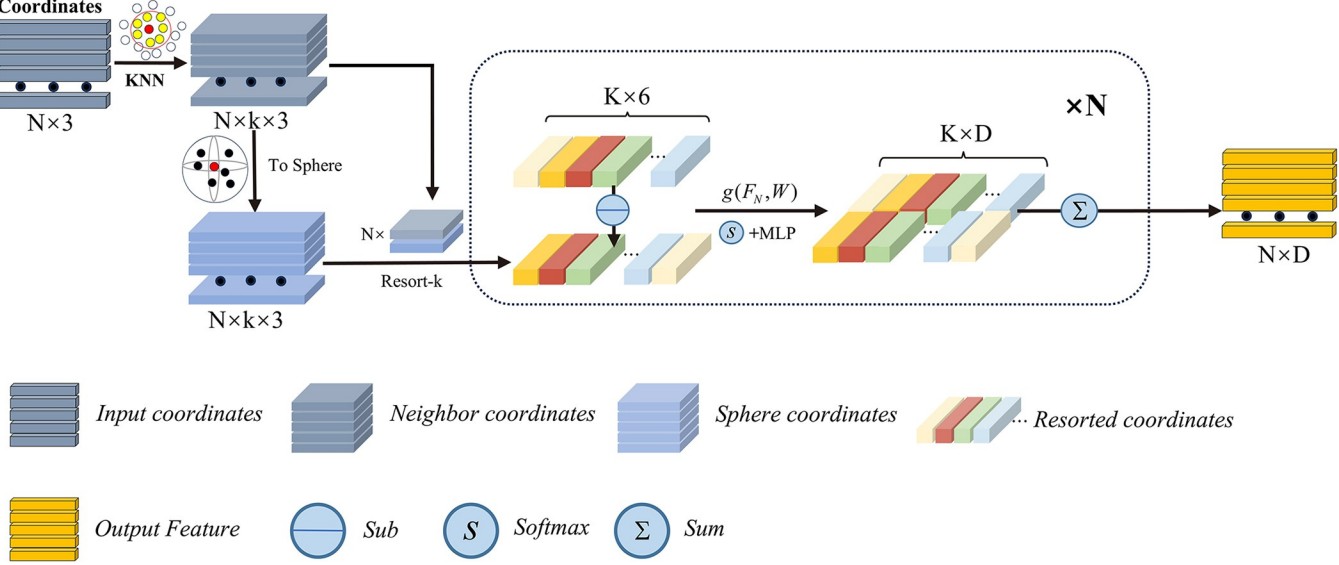

**Fig 2. PLAB structure, which concatenates the two coordinate representations into the local module after coordinate transformation, subtracts neighboring coordinates, and uses attentive pooling to weight and aggregate features.**

In local feature sorting and encoding, for the k-nearest points of each central point $P_i = \{P_i^1 \ldots P_i^k\}$, the Cartesian coordinate system is converted into a spherical coordinate system, and the adjacent points are sorted according to their polar angles to obtain the sorted adjacent points $\hat{P}_i = \{\hat{P}_i^1 \ldots \hat{P}_i^k\}$. The features of the adjacent point pairs are calculated using the sorted adjacent points, and the dimension is extended. This is defined as follows:

$$F_i^k = MLP\{\hat{F}c_i^k; \hat{F}s_i^k\}$$
$$\hat{F}c_i^k = Fc_i^k - Fc_i^{k-1}, \hat{F}s_i^k = Fs_i^k - Fs_i^{k-1}$$

(1)

where $Fc_i$ and $Fs_i$ represent the Cartesian coordinate-based $P_i$ features and spherical coordinate-based $\hat{P}_i$ features of neighboring points after sorting, respectively. The adoption of these two types of coordinate representations can mitigate the feature ambiguity caused by too close a proximity of neighboring points [29].

The attention score is then calculated. After obtaining the features of the local point pairs $F_i = \{F_i^1 \ldots F_i^k\}$, a set of shared functions $g()$ is used to obtain the weighted attention scores of the features for each feature channel. Finally, the *sum* function is used to aggregate the adjacent points. This is defined as follows:

$$F_i = \sum_{k=1}^{k} (F_i^k \cdot s_i^k)$$
$$s_i^k = g(\hat{F}_i^k, W)$$

(2)

where $W$ is the learnable weight of the shared MLP, and $g()$ is formed by combining the shared MLP with *softmax*. The feature aggregation of neighboring points using this method not only brings the advantage of order independence but also means the local features of neighboring points play a role together, similar to the encoding process of LBP.

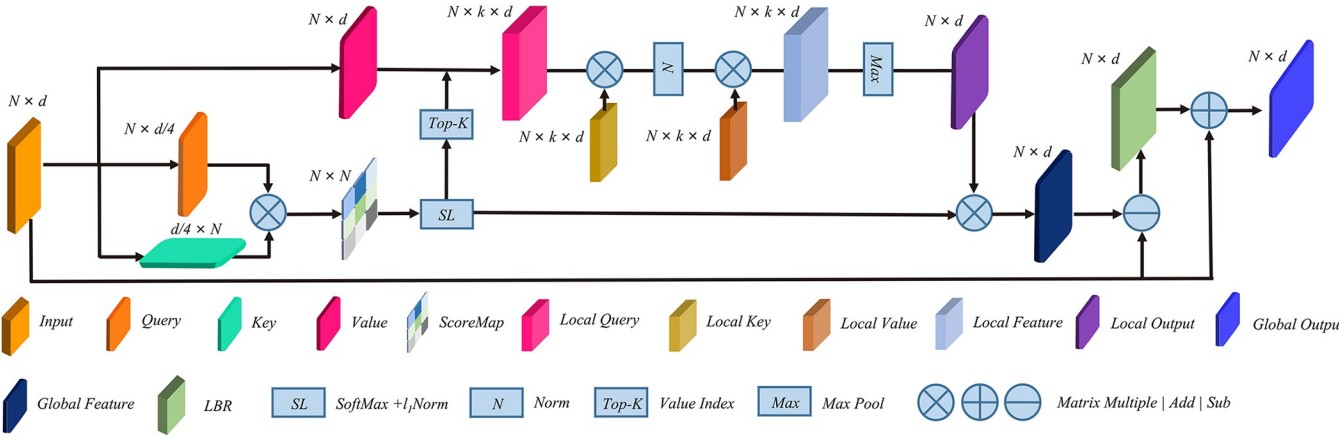

**Fig 3. DAL structure, which consists of offset-attention and external-attention.**

## Dual attention layer

As show in Fig 3, the DAL contains two types of attention mechanisms: Offset-Attention and External-Attention. Although these two attention mechanisms are based on existing models, they are combined by ingenious design to pay attention to the global and local relations at the same time. Inspired by DGCNN, the feature vector for local aggregation is formed by calculating the pairwise feature vector distance in the feature space and selecting the k-nearest neighbors. The main role of self-attention is to determine the relevance of each vector to the other vectors, and $W$ places an importance weight on each element. Instead of using the pairwise feature space distances, we utilize a dot product score matrix to select the feature vectors corresponding to the Top-K scores, thereby forming the feature vector for local aggregation. The local Transformer module employs External-Attention. In addition, the memory unit implicitly considers the relationship between different feature maps. When the input is local information, the memory unit considers the relationships among all local features.

N points are input with d-dimensional features from the PLAB layer. $Q$, $K$, and $V$ are obtained by $F_s \in \mathbb{R}^{N \times d}$ as the linear transformation of the *query*, *key*, and *value*, respectively:

$$(Q, K, V) = F_s \cdot (W_q, W_k, W_v)$$
$$Q, K \in \mathbb{R}^{N \times d/4}, V \in \mathbb{R}^{N \times d} \tag{3}$$

Where $W_q$, $W_k$ and $W_v$ are a shared learnable linear transformation, and the dimensions of *query* and *key* are reduced to *d/4* for efficient computation.

We compute the attention weights by matrix dot product using *query* and *key*:

$$\tilde{A} = (\tilde{a})_{i,j} = Q \cdot K^T \tag{4}$$

Where $\tilde{A}$ is the attention score matrix obtained by computing the dot product of $Q$ and $K$. Specifically, based on the query vector $Q$, a weight distribution $\tilde{A}$ is obtained by calculating the similarity of all query vectors $Q$ with all key vectors $K$.

We normalize the weights and assign them to the score matrix $\tilde{A}$ as follows:

$$(\bar{a})_{i,j} = softmax(\tilde{a}_{i,j}) = \frac{exp(\tilde{a}_{i,j})}{\sum\limits_{k} exp(\tilde{a}_{k,j}) + C}$$

$$A_g = a_{i,j} = \frac{\bar{a}_{i,j}}{\sum\limits_{k} \bar{a}_{i,k}} \tag{5}$$

where *softmax* enhances the difference between the values through exponential operations, so that larger attention scores are given higher weights in the probability distribution, and the influence of smaller attention scores is suppressed, and then $l_1$-*norm* is applied for further normalization to reduce the noise effect.

Next, we select the Top-K attention scores from matrix $A_g$ to obtain $V_k$, which corresponds to the selected $V$ values. $V_k$ is then used as input for the local module:

$$\tilde{V}_k = V - V_k \tag{6}$$

where $V_k$ is the nearest neighbor feature in feature space obtained by score matrix indexing, local features $\tilde{V}_k$ are obtained by establishing the relative relationship.

$$A_l = Norm(\tilde{V}_k M_k^T) \tag{7}$$

$$\bar{V}_k = A_l M_v \tag{8}$$

where $M_k$ and $M_v$ are learnable linear layers. Since $\tilde{V}_k$ corresponds to a neighborhood in the feature space and each feature vector affects $M_k$ and $M_v$, a memory cell is formed that takes into account all local features. After computing $A_l$, step (5) is still performed. Finally, $\bar{V}_k$ is the computed result of the local External-Attention. We obtain the local attention feature matrix, aggregate the features, and compute the global attention as follows:

$$\bar{V} = max(\bar{V}_k) \tag{9}$$

Where the local feature $\bar{V}_k$ is aggregated into feature $\bar{V}$ after maximum pooling.

$$F_{sa} = A_g \cdot \bar{V} \tag{10}$$

the local aggregated features are weighted by the attention score matrix $A_g$ to obtain the global self-attention $F_{sa}$.

The final output is obtained through the bias module:

$$F_{out} = BR(LBR(F_s - F_{sa}) + F_s) \tag{11}$$

Where *LBR* stands for *Linear*, *BatchNorm*, and *ReLU*. This part is used to enhance the model's feature learning.

## Experiments

After describing the methodology, we test the accuracy of the proposed method on the Model-Net40 dataset and the FG3D dataset for both coarse- and fine-grained classification tasks, and compare it with a range of classical and state-of-the-art models to illustrate the advancement of our method. In addition, we insert the PLAB module into other models to illustrate its effectiveness. We also provide implementation details of the models, including hardware configuration and hyperparameter settings.

## Experimental details

The classification network was implemented in PyTorch using an NVIDIA GEFORCE RTX 4060Ti GPU. We used the SGD optimizer with momentum 0.9, weight decay rate 0.0001, initialized learning rate set to 0.01, and cosine annealing schedule to adjust the learning rate of each epoch. We used 250 epochs for the classification network. The batch size was set to 24.

## Classification on ModelNet40 dataset

The ModelNet40 dataset contains 12311 CAD models with 40 classes, split it into 9843 training samples and 2468 test samples. Similar to most networks, we downsampled each input point cloud to 1024 points and used only 3D coordinates as input. The overall accuracy of each class was used for accuracy evaluation. The total number of Parameters and Floating-Point operations (FLOPs) were used for model size evaluation.

Table 1 lists the results for the various types of networks reproduced using our device. Our Transformer module and 3DGTN achieved the best results among all network results; however, our network only uses 3D coordinates, and the numbers of parameters and FLOPs are relatively low.

To verify the effectiveness of PLAB, it was incorporated into several classical models. To make the experiment more comprehensive, we fixed the random seed and ensured that other hyperparameters and the model network structure remained unchanged.

Table 2 shows that our method had improved effects on point-, graph-, and Transformer-based networks, as well as Fig 4 illustrates the reasons why PLAB can improve feature recognition. In PointNet++ and PCT, PLAB was inserted into the downsampling stage of the network, and the neighborhood size was set to that of the original network. In DGCNN, because Edge

**Table 1. Classification result on the ModelNet40 dataset.**

| Methods | Input | OA (%) | Parameters | FLOPs |
|---|---|---|---|---|
| Other Learning-based Methods | | | | |
| SO-Net [30] | 2048×3 | 90.9 | - | - |
| Point2Sequence [31] | 2048×3 | 92.6 | - | - |
| PointCNN [14] | 1024×3 | 91.7 | - | - |
| PointNet[†] [1] | 1024×3 | 89.2 | 3.47 M | 0.45 G |
| PointNet++(SSG) [†] [2] | 1024×3 | 92.4 | 1.48 M | 0.87 G |
| PointNet++(MSG) [†] [2] | 1024×3 | 92.7 | 1.75 M | 4.07 G |
| DGCNN[†] [17] | 1024×3 | 92.6 | 1.82 M | 2.43 G |
| DGCNN+Pnp-3D[†] [32] | 1024×3 | 92.5 | 1.93 M | 3.57 G |
| PointMLP[†] [33] | 1024×3 | 92.8 | 13.23 M | 15.73 G |
| DualMLP[†] [34] | 1024×3 | 93.1 | 14.32M | - |
| G-PointNet++[†] [35] | 1024×3 | 92.7 | - | - |
| Transformer-based Methods | | | | |
| GBNet[†] [36] | 1024×3 | 92.7 | 8.79 M | 9.86 G |
| Point Transformer[†] [37] | 1024×3 | 91.1 | 9.85 M | 18.40 G |
| PCT[†] [5] | 1024×3 | 92.4 | 2.88 M | 2.32 G |
| 3DGTN [†] [38] | 1024×3,N | **93.3** | 5.12 M | 3.09 G |
| DCNet[†] [39] | 1024×3 | 92.4 | 2.21 M | 7.80 G |
| PointConT[†] [40] | 1024×3 | 92.9 | - | - |
| Ours | 1024×3 | **93.3** | 2.43 M | 5.60 G |

† represents open source code recapitulation network experiments on NVIDIA GEFORCE RTX 4060Ti GPU. N a represents normal vector.

**Table 2. PLAB validity verification.**

| Based | Methods | OA(%) | +PLA(B%) |
|---|---|---|---|
| MLP | PointNet++(SSG)[†‡] [2] | 92.3 | 92.6(0.3↑) |
| | PointNet++(MSG) [†‡] [2] | 92.2 | 92.5(0.3↑) |
| Transformer | PCT[†‡] [5] | 92.1 | 92.7(0.6↑) |
| Graph | DGCNN[†‡] [17] | 92.4 | 92.7(0.3↑) |

† represents open source replicated network experiment and

‡ represents fixed random seed 1207.

Conv has the property of non-local aggregation, we only inserted PLAB after the first Edge Conv, and the other layers remained unchanged. Owing to the large scale of the MSG network, we set its batch size to 16, which resulted in the performance of MSG being lower than that of SSG. Additionally, fixing the random seed reduced the overall accuracy of the network by approximately 0.3 percentage points.

## Classification on FG3D dataset

The FG3D dataset is a fine-grained point cloud dataset that contains three major categories: aircraft, car, and chair, with 66 fine-grained subcategories and 25552 3D models. We used the same parameters as ModelNet40 for testing this dataset. The experimental results are shown in Table 3.

Table 3 shows the classification accuracies of airplane, car and chair using our method. DCNet performs better on fine-grained datasets compared to our method, and its model

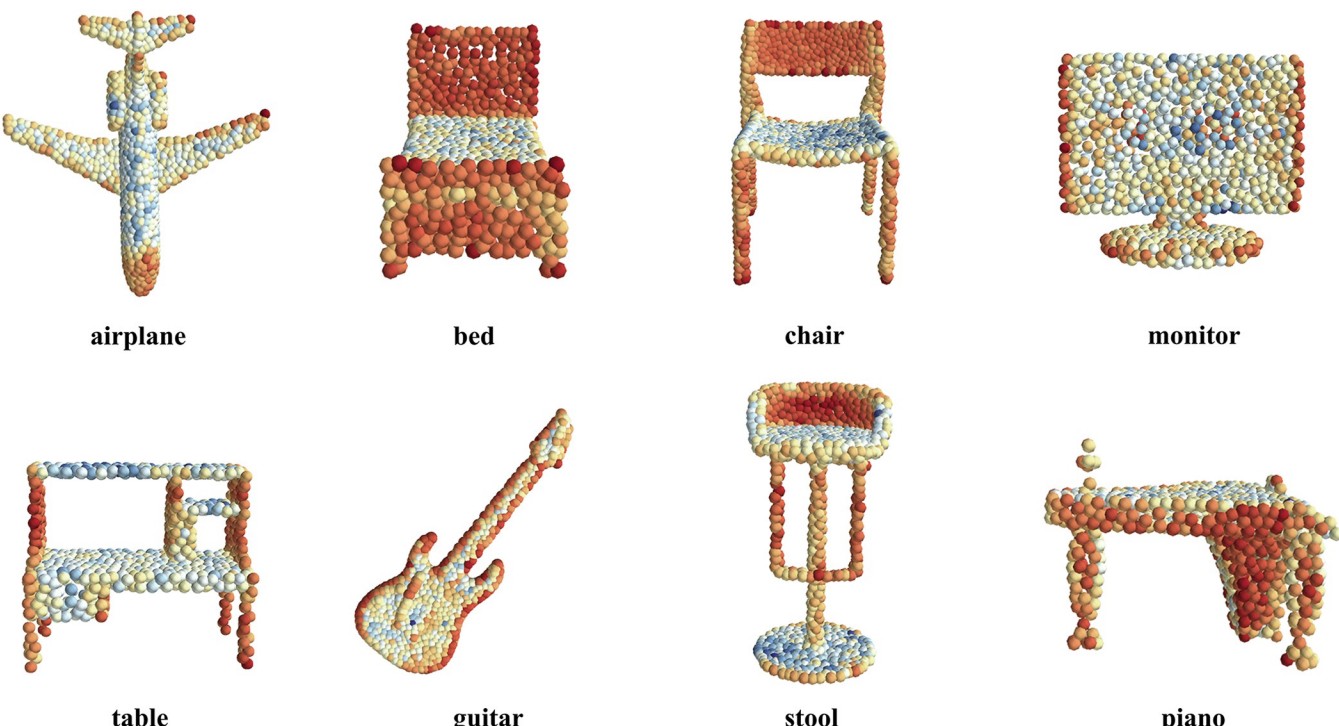

**airplane**          **bed**          **chair**          **monitor**

**table**          **guitar**          **stool**          **piano**

**Fig 4. Visualization of the normalized PLAB-encoded values.** It can be seen that different coding values are generated in the edge parts and different surfaces, which is key to assisting the classification task.

**Table 3. Classification results on the FG3D dataset.**

| Methods | Input | FG3D | | |
|---|---|---|---|---|
| | | Airplane | Car | Chair |
| MVCNN [8] | View 12×224² | 91.11 | 76.12 | 82.9 |
| FG3D-Net [41] | View 12×224² | 93.99 | 79.47 | 83.94 |
| SO-Net [30] | Point 2048×3 | 82.92 | 59.32 | 70.05 |
| Point2Sequence [31] | Point 2048×3 | 92.76 | 73.54 | 79.12 |
| PointCNN [14] | Point 1024×3 | 90.3 | 68.37 | 74.87 |
| PointNet† [1] | Point 1024×3 | 89.34 | 73 | 75.44 |
| PointNet++(MSG) † [2] | Point 1024×3 | 95.96 | 77.87 | 81.23 |
| MSP-Net† [42] | Point 1024×3 | 93.03 | 74.25 | 68.69 |
| PointAtrousGraph† [43] | Point 1024×3 | 95.22 | 74.77 | 79.2 |
| Point2SpatialCapsule† [44] | Point 1024×3 | 95.19 | 75.92 | 79.53 |
| DGCNN† [17] | Point 1024×3 | 93.6 | 72.1 | 79.53 |
| DGCNN+Pnp-3D† [32] | Point 1024×3 | 94.26 | 74.98 | 78.39 |
| GBNet† [36] | Point 1024×3 | 95.21 | 75.36 | 80 |
| Point Transformer† [37] | Point 1024×3 | 91.53 | 67.88 | 71.73 |
| PCT† [5] | Point 1024×3 | 95.16 | 78.89 | 81.37 |
| PointMLP† [33] | Point 1024×3 | 95.76 | 76.35 | 81.81 |
| DCNet† [39] | Point 1024×3 | 97.31 | 79.15 | 83.67 |
| Ours | Point 1024×3 | 96.02 | 76.73 | 81.33 |

† represents reproduction of open-source code experiments on an NVIDIA Tesla V100 GPU

focuses exclusively on local details because FG3D has greater intraclass sample variance, which requires that the model must be more attentive to the sample details, whereas ModelNet40 needs to be more attentive to the global features in order to achieve the interclass sample recognition, which also makes it perform poorly on coarse-grained datasets. Our method balances the two requirements well and achieves good results on both coarse- and fine-grained datasets, although it does not perform as well as DCNet on fine-grained datasets.

## Ablation study

To evaluate the effectiveness of each part of our method, we performed ablation experiments on the ModelNet40 dataset. In Table 4, use PCT as the baseline network, DAL-G represents the global part of DAL, and DAL-L represents the local part.

In addition, we experimented with the network accuracy effects of different network depths and neighborhood sizes, as shown in Table 5. The experimental results are consistent with the characteristics of the Transformer architecture [40].

**Table 4. Ablation study on model structure.**

| Baseline | PLAB | DAL-G | DAL-L | OA(%) | mAcc(%) |
|---|---|---|---|---|---|
| √ | | | | 92.4 | 89.4 |
| √ | √ | | | 92.6 | 89.9 |
| | √ | √ | | 92.7 | 90.1 |
| | √ | | √ | 92.6 | 89.9 |
| | | √ | √ | 93.1 | 90.5 |
| | √ | √ | √ | 93.3 | 91.1 |

**Table 5. Ablation study on different channel & stage and number of neighboring point.**

| Number of Neighbors(k) | OA(%) | mAcc(%) |
|---|---|---|
| 12 | 92.0 | 88.6 |
| 16 | 92.8 | 90.2 |
| 20 | 93.3 | 91.1 |
| 24 | 93.1 | 90.7 |
| 40 | 93.0 | 90.3 |
| Network depth (stage & channel) | OA(%) | mAcc(%) |
| 64-64-64 | 92.7 | 89.8 |
| 64-64-128-256 | 93.0 | 90.1 |
| 64-64-128-128-256 | 93.3 | 91.1 |
| 64-64-128-64-128-256 | 93.1 | 90.5 |

## Conclusion

In this paper, we proposed a simple and effective local complement module, PLAB, which is derived from the local binary pattern image texture extraction algorithm, to establish the relationship between neighborhood points and obtain effective features. Experiments showed that this module is suitable for different types of point cloud networks. In addition, we propose a dual local and global attention mechanism, DAL, which not only has the long-range dependence property of the Transformer structure but also pays attention to consistency in the feature space. The attention score map was used to obtain consistent features in the feature space for each point as the input of local attention. Considering the computing power consumed by the Transformer network, the lightweight attention mechanism of External-Attention was used to calculate the local feature attention in the local attention part. Finally, we proved the effectiveness of the proposed model on the ModelNet40 and FG3D datasets.

## Author Contributions

**Conceptualization:** Lei Wang.

**Data curation:** Dashi Qiu.

**Formal analysis:** Xingxing Xiao.

**Funding acquisition:** Zhenqing Yang.

**Investigation:** Dong Li.

**Methodology:** Lei Wang.

**Project administration:** Ming Huang.

**Resources:** Ming Huang.

**Software:** Ming Huang.

**Supervision:** Lei Wang.

**Validation:** Rui Wu.

**Visualization:** Dong Li.

**Writing – original draft:** Lei Wang.

**Writing – review & editing:** Lei Wang, Cai Chen.

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
