## [Decision Letter · Decision Letter 0]

16 Jun 2024

PONE-D-24-19727Learning features between neighboring points for point cloud classificationPLOS ONE

Dear Dr. wang,

Thank you for submitting your manuscript to PLOS ONE. After careful consideration, we feel that it has merit but does not fully meet PLOS ONE’s publication criteria as it currently stands. Therefore, we invite you to submit a revised version of the manuscript that addresses the points raised during the review process.

The paper presents the Point Cloud Local Auxiliary Block (PLAB) and the Dual Attention Layer (DAL), enhancing feature representation in point clouds. While the experimental results are promising, the manuscript requires significant revisions. The introduction should better connect existing methods to the authors' work, highlighting innovations. Additionally, the clarity of images and detailed explanations of formulas need improvement. Finally, more recent comparison methods should be included to strengthen the research context and relevance.

We look forward to receiving your revised manuscript.

Kind regards,

Ayesha Maqbool, PhD

Academic Editor

PLOS ONE

Journal Requirements:

"This study is sponsored by the BUCEA Doctor Graduate Scientific Research Ability Improvement Project(DG2024034) and National Natural Science Foundation of China (42171416)."

4. "Thank you for stating the following in the Acknowledgments Section of your manuscript: 

"This study is sponsored by the BUCEA Doctor Graduate Scientific Research Ability Improvement Project(DG2024034) and National Natural Science Foundation of China (42171416)."

"This study is sponsored by the BUCEA Doctor Graduate Scientific Research Ability Improvement Project(DG2024034) and National Natural Science Foundation of China (42171416)."

"NO authors have competing interests"

6. Please amend the manuscript submission data (via Edit Submission) to include authors Dr. Ming Huang, Dr. Rui Wu, Dr. Dashi Qiu, Dr. Xingxing Xiao, Dr. Dong Li, and Dr. Cai Chen.

7. Please amend your list of authors on the manuscript to ensure that each author is linked to an affiliation. Authors’ affiliations should reflect the institution where the work was done (if authors moved subsequently, you can also list the new affiliation stating “current affiliation:….” as necessary).

8. We note that Figure [1] includes an image of a [patient / participant / in the study]. 

Reviewers' comments:

Reviewer's Responses to Questions

**Comments to the Author**

1. Is the manuscript technically sound, and do the data support the conclusions?

Reviewer #1: No

Reviewer #2: Yes

2. Has the statistical analysis been performed appropriately and rigorously? 

Reviewer #1: No

Reviewer #2: Yes

3. Have the authors made all data underlying the findings in their manuscript fully available?

Reviewer #1: Yes

Reviewer #2: Yes

4. Is the manuscript presented in an intelligible fashion and written in standard English?

Reviewer #1: No

Reviewer #2: Yes

5. Review Comments to the Author

Reviewer #1: This paper introduces the Point Cloud Local Auxiliary Block (PLAB), inspired by local binary pattern algorithms, to improve neighborhood feature representation in point clouds. Additionally, the Dual Attention Layer (DAL), a Transformer structure, captures both local and global features using attention score maps. Experimental results show enhanced learning of features on various point cloud datasets. The authors will publish their code and experimental logs on GitHub, promoting transparency and reproducibility.

Most of the comparison methods are too outdated, especially other learning-based methods.

Why modelnet40 is conduct on 4060ti, fg3d is on 3060ti? Why not both using 4060ti?

In fg3d, why dcnet is better than the method proposed in this paper?

The references and comparison methods are too outdated. It is now 2024, so at least try to compare the methods from 2023 to the present.

Which benchmark network is this paper primarily based on?

There are many typos and grammar issues in the paper. For example, “Relu”; line 226 “… and …”;

There are some issues with the formatting of the paper. Eq. (4), (5) (8); The title and body of Table 1 are separated. I suggest the author use the latex template of PLOS ONE.

In table 1, the “point” in the table is redundant.

Most of the comparison methods are too outdated.

Reviewer #2: In this paper, the author proposed a simple and effective local complement module, PLAB, which is derived from the local binary pattern image texture extraction algorithm, to establish the relationship between neighborhood points and obtain effective features. The text performs well in the chart and experimental sections, but there are some minor issues in the introduction of the work.

Here are a few suggestions:

1.The percentage symbols in Equations 4, 5, 7, and 8 are misaligned. It is recommended to adjust the size of the percentage symbols to make them look more coordinated.

2.It is suggested that the author introduces more about the connection between different methods and their own work in the related work section. Explain how your method finds innovation among many methods, rather than merely listing the contents of others' work extensively.

3.It is recommended to provide more detailed explanations for the formulas in the text.

4.The images inserted in the text are mostly of insufficient clarity; it is recommended to replace them with higher-resolution images.

6. PLOS authors have the option to publish the peer review history of their article (what does this mean?). If published, this will include your full peer review and any attached files.

Reviewer #1: No

Reviewer #2: No

---

## [Author Response · Author response to Decision Letter 0]

30 Aug 2024

response to editor:

point 1:Please ensure that your manuscript meets PLOS ONE's style requirements, including those for file naming. 

response 1:We have carefully revised the manuscript according to the template, and if it still does not meet the requirements, we will continue to revise it.

point 2:Please note that PLOS ONE has specific guidelines on code sharing for submissions in which author-generated code underpins the findings in the manuscript. In these cases, we expect all author-generated code to be made available without restrictions upon publication of the work.

response 2:Our code is on other devices, and we hope to release it after the article is accepted.

point 3:Please state what role the funders took in the study. If the funders had no role, please state: "The funders had no role in study design, data collection and analysis, decision to publish, or preparation of the manuscript." If this statement is not correct you must amend it as needed. Please include this amended Role of Funder statement in your cover letter; we will change the online submission form on your behalf.

response 3:We stated "The funders had no role in study design, data collection and analysis, decision to publish, or preparation of the manuscript." in our cover letter.

point 4:We note that you have provided funding information that is not currently declared in your Funding Statement. However, funding information should not appear in the Acknowledgments section or other areas of your manuscript. We will only publish funding information present in the Funding Statement section of the online submission form. Please remove any funding-related text from the manuscript and let us know how you would like to update your Funding Statement.

response 4:We have removed the mention of fund support in the acknowledgements, so please update the fund support "BUCEA Doctor Graduate Scientific Research Ability Improvement Project(DG2024034) and National Natural Science Foundation of China (42171416).".

point 5:Please complete your Competing Interests on the online submission form to state any Competing Interests. If you have no competing interests, please state "The authors have declared that no competing interests exist."

response 5:We have already noted "The authors have declared that no competing interests exist." in the cover letter.

Point 6: Please amend the manuscript submission data (via Edit Submission) to include authors Dr. Ming Huang, Dr. Rui Wu, Dr. Dashi Qiu, Dr. Xingxing Xiao, Dr. Dong Li, and Dr. Cai Chen.

Response 6: We will amend the manuscript submission data.

Point 7: Please amend your list of authors on the manuscript to ensure that each author is linked to an affiliation. Authors’ affiliations should reflect the institution where the work was done (if authors moved subsequently, you can also list the new affiliation stating “current affiliation:….” as necessary).

Response 7:We have amend list of authors on the manuscript.

Point 8: We note that Figure [1] includes an image of a [patient / participant / in the study]. If you are unable to obtain consent from the subject of the photograph, you will need to remove the figure and any other textual identifying information or case descriptions for this individual.

Response 8: We were unable to obtain consent, so we removed Figure 1.

Response to Reviewer#1:

Point 1. Most of the comparison methods are too outdated, especially other learning-based methods.

Response 1: Thank you for your suggestion. In fact, the writing of this manuscript was completed at the end of 2023, and the methods we compared included some classic papers as well as newer papers from 2022 and 2023, such as 3DPCT and PointConT. However, we agree with your suggestion and add some of the latest methods for comparison. Specifically refer to lines 159-165 and Table 1 of the Revised Manuscript with Track change.

Point 2. Why modelnet40 is conduct on 4060ti, fg3d is on 3060ti? Why not both using 4060ti?

Response 2: Thank you for your careful review. Since the experiments were conducted on different devices, we have repeated the experiments on the 4060Ti device. Specifically refer to Table 3 of the Revised Manuscript with Track change.

Point 3. In fg3d, why dcnet is better than the method proposed in this paper?

Response 3：Your question is well-taken. After extensive experiments, we found that many existing models perform differently on different datasets. For example, while DCNet outperformed our proposed method on the FG3D dataset, it only achieved a 92.4% overall accuracy on the ModelNet40 dataset, significantly lower than our 93.3% overall accuracy. Furthermore, while our proposed method's performance on the FG3D dataset was not the best, its results were still competitive, indicating that our method is adaptable to different datasets.

Point 4. The references and comparison methods are too outdated. It is now 2024, so at least try to compare the methods from 2023 to the present.

Response 4：Thank you for your suggestion. As mentioned in Point 1, the methods we compared include both classic and newer ones, and we have added new methods to the manuscript.

Point 5. Which benchmark network is this paper primarily based on?

Response 5：Thank you for your question. Our benchmark network is PCT, and we highlighted the issue of this oversight in the manuscript. Specifically refer to lines 403 of the Revised Manuscript with Track change.

Point 6. There are many typos and grammar issues in the paper. For example, “Relu”; line 226 “… and …”;

Response 6：Thank you for your suggestion. We carefully rechecked the manuscript and made corrections to spelling and grammar. Specifically refer to the Revised Manuscript with Track change.

Point 7. There are some issues with the formatting of the paper. Eq. (4), (5) (8); The title and body of Table 1 are separated. I suggest the author use the latex template of PLOS ONE.

Response 7：We have modified the paper according to the formatting requirements of PLOS ONE and reorganized the equations. Specifically refer to the Revised Manuscript with Track change.

Point 8. In table 1, the “point” in the table is redundant.

Response 8："point" was redundant, and we have made the correction in the table 1.

Point 9. Most of the comparison methods are too outdated.

Response 9：Thank you for your suggestion. In order to compare newer methods, we have added newer methods in the Related Work section, and conducted an experimental comparison of the latest method, which is shown in Table 1.

Response to Reviewer#2:

Point 1. The percentage symbols in Equations 4, 5, 7, and 8 are misaligned. It is recommended to adjust the size of the percentage symbols to make them look more coordinated.

Response 1：We revised the formula expression to make it clearer and more aesthetically pleasing. Specifically refer to the Revised Manuscript.

Point 2. It is suggested that the author introduces more about the connection between different methods and their own work in the related work section. Explain how your method finds innovation among many methods, rather than merely listing the contents of others' work extensively.

Response 2：Thank you for your suggestion. We have added content to enhance the relevance of the related work to this Manuscript, to highlight its relevance. Specifically refer to lines 127-133,171-174,181-183 of the Revised Manuscript with Track change.

Point 3. It is recommended to provide more detailed explanations for the formulas in the text.

Response 3：We have provided a more detailed explanation of the formulas in the Manuscript. Specifically refer to lines 289-330 of the Revised Manuscript with Track change.

Point 4. The images inserted in the text are mostly of insufficient clarity; it is recommended to replace them with higher-resolution images.

Response 4：Thank you for your suggestion. We have modified some unclear images and tried to make them more visually appealing, as shown in Fig 4. We have also removed Fig 1 due to copyright issues. Specifically refer to line 383 of the Revised Manuscript with Track change.

---

## [Decision Letter · Decision Letter 1]

3 Oct 2024

PONE-D-24-19727R1Learning features between neighboring points for point cloud classificationPLOS ONE

Dear Dr. wang,

Thank you for submitting your manuscript to PLOS ONE. After careful consideration, we feel that it has merit but does not fully meet PLOS ONE’s publication criteria as it currently stands. Therefore, we invite you to submit a revised version of the manuscript that addresses the points raised during the review process.

The article "Learning Features Between Neighboring Points for Point Cloud Classification" presents a valuable contribution; however, to enhance its quality, I recommend improving the analysis section and presenting the findings in greater depth. The objectives of the work should be further elaborated, with a clearer explanation of how these objectives are achieved. Providing a more structured presentation of the alignment between the research goals and results will significantly strengthen the paper.

We look forward to receiving your revised manuscript.

Kind regards,

Ayesha Maqbool, PhD

Academic Editor

PLOS ONE

Journal Requirements:

Reviewers' comments:

Reviewer's Responses to Questions

**Comments to the Author**

1. If the authors have adequately addressed your comments raised in a previous round of review and you feel that this manuscript is now acceptable for publication, you may indicate that here to bypass the “Comments to the Author” section, enter your conflict of interest statement in the “Confidential to Editor” section, and submit your "Accept" recommendation.

Reviewer #1: (No Response)

Reviewer #2: (No Response)

2. Is the manuscript technically sound, and do the data support the conclusions?

Reviewer #1: Partly

Reviewer #2: Partly

3. Has the statistical analysis been performed appropriately and rigorously? 

Reviewer #1: Yes

Reviewer #2: N/A

4. Have the authors made all data underlying the findings in their manuscript fully available?

Reviewer #1: Yes

Reviewer #2: No

5. Is the manuscript presented in an intelligible fashion and written in standard English?

Reviewer #1: Yes

Reviewer #2: No

6. Review Comments to the Author

Reviewer #1: Most of comments have been addressed by the authors. However, all the figures in this paper look very rough, the images look very blurry. It is recommended to use vector graphics instead.

Another concern is that most of the network figures look very similar to the figures in the publised papers, so the novely of this paper also need to be concerned.

Reviewer #2: 1.There are several spelling and grammar errors in the text, such as the phrase "performed we this module" in the abstract. These details need to be meticulously proofread to prevent unnecessary mistakes that could impact the quality of the paper.

2.It has been noted that the images in the paper, particularly Figure 4, lack clarity. It is recommended to ensure that all images are of high resolution to facilitate easier viewing for readers.

3.In Table 1, the title and body are misaligned in several areas, which affects readability. Furthermore, the term "point" is redundant and should be eliminated from the table.

4.The explanations for formulas 4, 5, 7, and 8 lack sufficient detail. It is recommended to provide more comprehensive explanations of the derivation and underlying principles of these formulas. This will help ensure that readers can clearly grasp the derivation process and the physical significance of each step.

5.Although the "Related Work" section references relevant studies, the analysis primarily consists of a basic enumeration of their contents. It is advisable to emphasize the distinctions and innovations of your method in comparison to existing approaches to further strengthen the paper's persuasiveness.

6.Although the experimental results indicate that the method performs well on certain datasets, a more in-depth discussion is necessary regarding specific cases (e.g., the reasons why the method underperforms compared to DCNet on the FG3D dataset). Clarifying these differences will enhance readers' understanding of the method's strengths and weaknesses.

7.Some paragraphs transition rather abruptly, particularly between the introduction of the methods and the experiments section. Incorporating transitional sentences can enhance the overall flow and coherence of the paper's structure.

7. PLOS authors have the option to publish the peer review history of their article (what does this mean?). If published, this will include your full peer review and any attached files.

Reviewer #1: No

Reviewer #2: No

---

## [Author Response · Author response to Decision Letter 1]

1 Nov 2024

response to review #1:

Thanks to your suggestions, we have replaced all the images with higher resolution images and Figure 4 has been redrawn with new tools to make it look better. You can view it below or in Revised Manuscript with Track change.For problems where the numbers are very close, because the classification task is close to saturation, even small overall accuracy gains still require quite complex improvements, and from the published networks, each percentage point of overall accuracy gain for the classification task requires approximately one to two years of experimentation by the relevant researchers around the world to improve the network performance.

response to review #2:Detailed responses have been put into the “Response to Reviewer” file.

Response 1：Thanks to your suggestion, we have carefully checked the whole text and corrected several grammatical and spelling errors. You can view in Revised Manuscript with Track change.

Response 2：Thanks to your suggestions, we have replaced all the images with higher resolution images and Figure 4 has been redrawn with new tools to make it look better. You can view it in Revised Manuscript with Track change.

Response 3：Thanks to your careful scrutiny, we've reworked the problems with the table 1. You can check it out in the Revised Manuscript.

Response 4：Thanks for your suggestion, I have provided more explanations for formulas 4, 5, 7, and 8. You can check it out in the Revised Manuscript.

Response 5：Thanks to your suggestion, we've added more differentiating descriptions to our related work to highlight the distinctiveness and innovation of our approach. You can check it out in the Revised Manuscript.

Response 6：Thank you for your insightful suggestions. Indeed, networks targeted for fine-grained point cloud classification tasks differ in their design goals from those for traditional point cloud classification tasks. For the traditional point cloud dataset (ModelNet40), the network usually needs to learn the global structure of the point cloud samples in order to perform classification efficiently. However, since fine-grained point cloud datasets (FG3D) have relatively small interclass sample differences and relatively large intraclass sample differences, this property motivates the network to have to pay more attention to the local detailed features in order to achieve more accurate classification. fine-grained point cloud classification differs significantly from traditional point cloud classification in terms of the focus of feature learning. In order to focus on the ModelNet40 dataset, our proposed network concentrates on learning the global structure of the point cloud samples for the classification task, and thus has a lower accuracy than DCNet on the FG3D dataset. However, it is worth noting that our net exhibits superior performance on the fine-grained dataset FG3D compared to existing networks such as PCT. This is mainly due to the innovative design of the dual attention layer in terms of local feature capture, which enables the network to extract some fine features more efficiently, thus jointly improving the coarse- and fine-grained classification accuracy. We also analyse this problem in the manuscript，you can check it out in the Revised Manuscript.

Response 7：The excesses of these two parts were indeed abrupt, we readjusted the transitional sentences to make the manuscript smoother. You can check it out in the Revised Manuscript.

---

## [Editor Report · Decision Letter 2]

5 Nov 2024

Learning features between neighboring points for point cloud classification

PONE-D-24-19727R2

Dear Dr. Wang,

We’re pleased to inform you that your manuscript has been judged scientifically suitable for publication and will be formally accepted for publication once it meets all outstanding technical requirements.

Kind regards,

Ayesha Maqbool, PhD

Academic Editor

PLOS ONE
---

## [Editor Report · Acceptance letter]

10 Dec 2024

PONE-D-24-19727R2 

PLOS ONE

Dear Dr. Wang, 

I'm pleased to inform you that your manuscript has been deemed suitable for publication in PLOS ONE. Congratulations! Your manuscript is now being handed over to our production team.

Kind regards, 

on behalf of

Dr. Ayesha Maqbool 

Academic Editor

PLOS ONE